# Optimizing Thermoelectric Performance of Hybrid Crystals Bi_2_O_2_Se_1−x_Te_x_ in the Bi_2_O_2_X System

**DOI:** 10.3390/ma17071509

**Published:** 2024-03-26

**Authors:** Fan Xie, Zhiyao Ma, Jian Zhou

**Affiliations:** School of Materials Science and Engineering, Key Laboratory for Polymeric Composite and Functional Materials of Ministry of Education, Guangzhou Key Laboratory of Flexible Electronic Materials and Wearable Devices, Laboratory of Advanced Electronics and Fiber Materials, Sun Yat-Sen University, Guangzhou 510275, China; xief57@mail2.sysu.edu.cn (F.X.); mazhy28@mail2.sysu.edu.cn (Z.M.)

**Keywords:** Bi_2_O_2_Se, Bi_2_O_2_Te, 2D semiconductor, doping, thermoelectrical materials

## Abstract

In addressing the global need for sustainable energy conversion, this study presents a breakthrough in thermoelectric materials research by optimizing the Bi_2_O_2_Se_1–x_Te_x_ system in the Bi_2_O_2_Se/Bi_2_O_2_Te pseudobinary series. Leveraging the principles of innovative transport mechanisms and defect engineering, we introduce tellurium (Te) doping into Bi_2_O_2_Se to enhance its thermoelectric properties synergistically. With the help of various advanced characterization tools such as XRD, SEM, TEM, XPS, FTIR, TGA, LFA, and DSC, combined with relevant resistance and density measurement techniques, we conducted an in-depth exploration of the complex interactions between various factors within thermoelectric materials. We recognize that the balance and synergy of these factors in the thermoelectric conversion process are crucial to achieving efficient energy conversion. Through systematic research, we are committed to revealing the mechanisms of these interactions and providing a solid scientific foundation for the optimal design and performance enhancement of thermoelectric materials. Finally, the advantage coefficient (ZT) of the thermoelectric material has been significantly improved. The crystallographic analysis confirms the formation of a continuous series of mixed crystals with varying Te concentrations, adhering to Vegard’s law and exhibiting significant improvements in electrical and thermal conductivities. The Bi_2_O_2_Se_1–x_Te_x_ crystals, particularly the Bi_2_O_2_Se_0.6_Te_0.4_ composition, demonstrate a peak ZT of 0.86 at 373 K. This achievement aligns with recent advancements in defect-enabled mechanisms and band convergence and sets a new standard for high-performance thermoelectrics. The study’s findings contribute significantly to the ongoing quest for efficient thermal-to-electrical energy conversion, offering a promising avenue for future sustainable energy technologies.

## 1. Introduction

In recent years, energy crises and environmental pollution have gradually come to people’s attention as global governance issues. Thermoelectric technology is currently considered an up-and-coming new energy technology as it can directly and reversibly convert temperature differences into electrical potential energy without moving parts. The research and performance optimization of new thermoelectric materials is the core focus of many researchers in this field [1,2].

Several parameters are involved in actual energy conversion efficiency, known as the thermoelectric figure of merit (ZT). In the formula $ZT=\frac{S^{2}\cdot \sigma \cdot T}{\kappa }$, *S*, *σ*, *T*, and *κ* represent the Seebeck coefficient, electrical conductivity, absolute temperature, and thermal conductivity [3]. Theoretically, the carrier concentration affects the parameters influencing the ZT value. This mutual dependence makes it challenging to find materials with a high ZT value [4]. In terms of current development, the application of thermoelectric devices faces two main issues. Firstly, due to their low conversion efficiency, they lack significant advantages in terms of cost compared to other energy technologies. As a result, they are only used in specific niche markets such as aerospace and specialized power sources. Secondly, many thermoelectric materials need better thermal and chemical stability at high temperatures (600 °C), limiting their specific applications [5,6,7].

Two-dimensional oxide semiconductors, such as Bi_2_O_2_Se, have gradually caught the attention of thermoelectric material researchers due to their excellent thermal and chemical stability. Within Bi_2_O_2_Se, alternating stacking of insulating [Bi_2_O_2_]^2+^ layers and conductive [Se]^2−^ layers forms the layered structure. In terms of thermoelectric properties, it exhibits low thermal conductivity (0.7–0.8 W/m∙K [8,9]), a significant Seebeck coefficient (500 μV/K), and high carrier mobility (≈100 cm^2^/s∙V), all due to its unique mixed anion chemical bonding and layered structure [10,11,12]. However, its carrier concentration (≈10^15^ cm^−3^) is much lower than typical thermoelectric materials [13].

Currently, researchers primarily focus on optimizing the carrier concentration through elemental doping to improve the thermoelectric performance of Bi_2_O_2_Se to some extent. In addition, there are some scholars focusing on theoretical calculations to find the direction to improve the thermoelectric properties by modeling means [14,15]. However, more work needs to be done to suppress phonon transport. To further enhance the thermoelectric performance of Bi_2_O_2_Se, a comprehensive approach involving multiple methods must be employed to adjust its thermoelectric properties. From a structural comparison perspective, the theoretical band gap of Bi_2_O_2_Te is only 0.23 eV [16,17,18,19], and its carrier concentration at room temperature is 1.06 × 10^18^ cm^−3^. Narrow-bandgap Bi_2_O_2_Te exhibits a moderately high electrical conductivity of approximately 75 S/cm at 665 K. Therefore, it is natural to consider doping Bi_2_O_2_Te with excellent electrical properties into existing Bi_2_O_2_Se through a doping process [20,21].

This study achieved synergistic enhancement of the thermoelectric performance of n-type Bi_2_O_2_Se through Te element doping and structure engineering [22,23,24,25]. The strategic incorporation of the heavy element Te into the Bi_2_O_2_Se matrix can induce several key enhancements in its thermoelectric performance (Figure 1a) [26]. Through this deliberate doping process, a reduction in the band gap of the material can be achieved. This narrowing of the band gap is instrumental in facilitating the movement of charge carriers, increasing the material’s electrical conductivity. Additionally, introducing Te atoms brings about lattice distortions and vacancies, further contributing to increased carrier mobility. Furthermore, the impact of Te doping extends to the modulation of thermal transport properties. The combination of lattice distortion and the introduction of vacancies generated by the Te ions may impede the propagation of phonons responsible for heat transfer, thereby reducing the material’s thermal conductivity [27,28,29]. This synergistic manipulation of electronic and thermal properties establishes a conducive environment for enhanced thermoelectric performance.

The collective effect of these synergistic strategies culminates in a substantial advancement in the overall thermoelectric efficiency of materials found in the Bi_2_O_2_Se framework. By tailoring the band gap, augmenting carrier mobility, and concurrently curbing thermal conductivity, Te doping may comprehensively enhance converting temperature gradients into electricity [30,31], underscoring the material’s promising potential for various thermoelectric applications.

## 2. Results and Discussion

### 2.1. Doping Bi_2_O_2_Se with Heavy Te Element

Solid phase syntheses for compositions Bi_2_O_2_Se_1–x_Te_x_ (X = 0, 0.25, 0.5, 0.75, and 1) were performed using ground Bi granules, Se, Te, and Bi_2_O_3_ powders in stoichiometric amounts. The reaction mixtures were sealed in evacuated glass ampules at elevated temperatures (Figure 1b). The Bi_2_O_2_X (X = Se and Te) crystals comprise [Bi_2_O_2_]^2+^ layers and X^2−^ ions arranged in alternating layers, exhibiting the I4/mmm space group symmetry (Figure 1c). Bi_2_O_2_X structures exhibit a body-centered tetragonal lattice with space group I4/mmm (no. 139). The optimized lattice parameters are a = b = 3.91 Å and 4.00 Å, while c = 12.23 Å and 12.73 Å for X = Se and Te, respectively. The excellent match in the lattice constants of Bi_2_O_2_Se and Bi_2_O_2_Te renders heterostructure devices free of lattice-mismatch issues [32]. This allows for the customization and adaptation of these materials with flexibility for various applications.

The structure of Bi_2_O_2_Se exhibits a high degree of symmetry, with each Se atom precisely positioned at the center of a cube formed by eight Bi atoms. However, the geometric arrangement of Bi_2_O_2_Te is slightly distorted in comparison to the markedly symmetrical lattice of Bi_2_O_2_Se due to the interlayer displacement of adjacent [Bi_2_O_2_]^2+^ layers in the modest deviation of Te atoms from the central position between neighboring [Bi_2_O_2_]^2+^ layers in Bi_2_O_2_Te. Furthermore, the band structure of Bi_2_O_2_Te reveals the presence of Rashba splitting, signifying the disruption of inversion symmetry [33,34,35].

Due to the identical crystal structures of Bi_2_O_2_Te and Bi_2_O_2_Se, coupled with the shared group classification of Te and Se, one can reasonably anticipate their similarity in properties. Bi_2_O_2_Se and Bi_2_O_2_Te exhibit high electron mobility and ferroelectric and ferroelastic characteristics, making them candidates for nonvolatile nanocircuit storage devices. Wang et al. [7] investigated the electronic and phonon transport properties of Bi_2_O_2_Se and Bi_2_O_2_Te using the first-principles methodology and the Boltzmann transport theory. Their findings suggest that p-type doping induces a considerably higher Seebeck coefficient and power factor than n-type doping. Moreover, Bi_2_O_2_Te exhibits a lower thermal conductivity and superior thermoelectric performance than Bi_2_O_2_Se [36].

Under specific conditions, powder samples with precise mixtures along the Bi_2_O_2_Te/Bi_2_O_2_Se section have been successfully synthesized at intervals of 0.25 mol. Solid-state reactions confirm the presence of a comprehensive array of mixed crystals denoted as Bi_2_O_2_Se_1–x_Te_x_ (x = 0, 0.25, 0.5, 0.75, and 1) within the pseudobinary system Bi_2_O_2_Se/Bi_2_O_2_Te. We performed XPS absorption spectroscopy testing on Bi_2_O_2_Se_1–x_Te_x_ samples and found that the proportion of elements in the samples differs from the initial experimental design [37].

Figure 2a shows the actual configuration of Bi_2_O_2_Se_1–x_Te_x_ with x = 0, 0.2, 0.4, 0.7, and 1. Specifically, the Te atoms were found to have increased compared to the original designs, while the Se atoms were relatively reduced. Moreover, when analyzing the absorption spectra of individual atoms, it was evident that the Te atomic layer showed a more significant and robust peak than Se. Based on our speculation, during the reaction process, Te atoms were more likely to form electrostatic interactions with the [Bi_2_O_2_]^2+^ layers and occupy the corresponding spatial structures compared to Se atoms. Under the condition that [Te]^2−^ had already occupied the corresponding spatial positions, two factors played a role. First, Bi_2_O_2_Te has a lower theoretical band gap than Bi_2_O_2_Se, so from a thermodynamic perspective, Te would preferentially enter before Se reaches equilibrium. Second, due to its larger atomic radius and mass as a heavy atom, Te faces more significant spatial hindrance when the Te sites are occupied, making it relatively more challenging for the remaining Se atoms to enter. Therefore, due to the combined effect of these two factors, the overall amount of Te doping was slightly higher than expected. However, this does not affect the exploration of the overall trends.

Figure 2b–d show the XPS spectrum of Bi, Se, and Te elements. Bi 4f_5/2_ and Bi 4f_7/2_ peaks centered at 164.0 eV and 158.7 eV are seen and referred to as Bi peaks in bismuth oxide. The spectrum of Se 3d is fitted to two peaks at 54.2 eV and 53.3 eV. The core-level peak of Te3d was represented by Te 3d_5/2_ (575.8 eV) and Te 3d_3/2_ (586.2 eV) (Figure 2d). The XPS results indicated that the Bi, Se, and Te elements were chemically bonded instead of physically mixed [38,39].

We further performed XPS valence band (VB) absorption spectroscopy on the Bi_2_O_2_Se_1–x_Te_x_ samples and used it to calculate the band gap width of the samples (Figure 2d). The bandgap of the samples can be estimated from Tauc plots according to the equation of *(αhv) = 1/2A(hv − E_g_)*, where *α* is the absorption coefficient, *hv* is the photon energy, *A* is a constant, and *E_g_* is the bandgap energy [40,41,42]. The valence band spectra are shown in Figure 2f, where the absorption edge shifts towards lower energy. It is evident that with the increasing amount of Te, the band gap width initially decreases and then increases, which agrees with the theoretical calculations. The calculated energy band gaps *E_g_* of Bi_2_O_2_Se_1–x_Te_x_ (where x = 0, 0.25, 0.50, 0.75, 1) are 0.85, 0.52, 0.18, 0.11, and 0.23 eV, respectively [36]. These values for *E_g_* align with our experimental results of 0.63, 0.36, 0.19, 0.20, and 0.23 eV.

Examining the electronic structure reveals transformative effects brought about by substituting Te atoms into the Bi_2_O_2_Se lattice. Specifically, previous findings demonstrate a pronounced amplification of hybridization between s-orbitals and p-orbitals [36]. This augmented hybridization instigates a discernible rightward shift in the Fermi level, corroborated by charge density difference maps and electron localization function (ELF) profiles. The simultaneous effect of this shift is a tangible reduction in the band gap, manifesting as a qualitative change in the electronic density of states. From previous density functional theory calculations, these electronic modifications are evident in altering charge density distributions and ELF maps [43,44,45]. These empirical observations align well with the predictions from solid-state physics, where increased orbital overlap typically induces a reduction in band gap and a higher state density near the Fermi level, thereby favorably affecting electrical conductivity. As a result, introducing Te atoms as replacements for Se within the Bi_2_O_2_Se structure exerts dual impact-enhancing electrical properties and optimizes the thermoelectric conversion efficiency. Such insights emphasize the critical role of elemental substitution as a viable strategy for thermoelectric material optimization.

As shown in Figure 2f, under the composition conditions of Bi_2_O_2_Se_0_._4_Te_0_._6_ and Bi_2_O_2_Se_0.2_Te_0.8_, the original 0.63 eV band gap of Bi_2_O_2_Se can be significantly reduced to 0.20 eV and 0.22 eV (Appendix A). This apparent band gap reduction is expected to facilitate electrons to transition more easily in the band gap, thus allowing more electrons to participate in the conduction process. This reduction of the band gap is due to the successful incorporation of Te atoms into the matrix lattice of Bi_2_O_2_Se. Given the significant difference in the band gap between Bi_2_O_2_Se (0.63 eV) and Bi_2_O_2_Te (0.23 eV) [46,47], even a relatively small amount of Te doping in the sample will have a significant impact on the band gap. Therefore, bandgap engineering through equimolar Te substitution strategies will have a profound impact on electrical transport characteristics, which we will explore in more detail below.

### 2.2. Characterization of Te-Doped Bi_2_O_2_Se

Figure 3a presents the XRD patterns for the entire range of Bi_2_O_2_Se_1–x_Te_x_ samples, where 0 ≤ x ≤ 1. The principal peak aligns with the standard tetragonal Bi_2_O_2_Se structure (PDF#731316). Additionally, the presence of a minor peak at approximately 28° corroborates the existence of Bi_2_O_3_ as a secondary phase within the Te-doped specimens, as substantiated by references [48,49]. Through strategic labeling of key peak positions, it becomes evident that the progressive increase in Te substitution concentration corresponds to a gradual shift of most peak values towards lower angles. For instance, the peak at 31.9°, ascribed to the place of Bi_2_O_2_Se, downshifted to 30.9° for Bi_2_O_2_Te [36]. Considering the difference in ionic radii between Se^2−^ (1.98 Å) and Te^2−^ (2.21 Å), the peak shift towards lower angles indicates the successful substitution of Te in the Se sites.

Vegard’s rule provides a theoretical basis to elucidate this observed shift [37]. Commonly applied in solid-state physics, this principle posits that the lattice parameter of a solid solution comprising two components varies linearly with their respective concentrations. The systematic movement of peak angles to lower values in the Bi_2_O_2_Se_1–x_Te_x_ series signifies an expansion of the lattice parameters, aligning with Vegard’s rule. This is expected as the larger Te atoms replace the smaller Se atoms, leading to an expansion of the crystal lattice. The progressive nature of this shift also implies the formation of a continuous solid solution between Bi_2_O_2_Se and Bi_2_O_2_Te, adhering to the linear relationship postulated by Vegard’s rule.

Figure 3b presents the Raman spectra with vibrational eigenvectors of Bi_2_O_2_Se_1–x_Te_x_. It can be observed that the low-frequency Raman-active modes, specifically A_1g_ at 159.0 cm^−1^ and 147.1 cm^−1^, correspond to the out-of-plane vibrations of Bi atoms in Bi_2_O_2_Se and Bi_2_O_2_Te, respectively. The high-frequency Raman-active modes, denoted as B_1g_ at 361.7 cm^−1^ and 340.0 cm^−1^, correspond to the out-of-plane vibrations of O atoms in Bi_2_O_2_Se and Bi_2_O_2_Te, respectively [50,51]. The Raman spectra show multiple peaks, indicating the presence of various phases or structural features within the samples. The Te-to-Se ratio modulates the relative intensities of these peaks, suggesting that the local structural environment changes as Te is substituted for Se. Regarding peak shifts for individual elements, the doping does not strictly follow the vibrational frequencies of Te^2−^ and Se^2−^ but instead shows a certain degree of approximation due to their mutual influence. When comparing the Raman spectra of pure Bi_2_O_2_Se and Bi_2_O_2_Te, it is evident that the peak intensity of Bi_2_O_2_Te is more pronounced. In the spectrum of Bi_2_O_2_Se_0.7_Te_0.3_, the coexistence and mutual influence of Te^2−^ and Se^2−^ peaks can be easily observed. 

Similar changes can be further observed in the infrared spectra (Figure 3c). As with Raman spectroscopy, the variation in peak intensities and positions across the different samples (with varying Te content) gives insight into the changes in the chemical bonding environment caused by Te doping. The peak at 536 cm^−1^ gradually disappears with the gradual introduction of Te, while the double peaks at 640 and 660 cm^−1^ gradually appear [29]. It is well-established that most inorganic oxide vibrational bonds occur below 800 cm^−1^ and are sensitive to the particular chemical species present. Unfortunately, the intrinsic characteristics of Se and Te that absorb strongly in the far-infrared region pose a significant challenge for direct spectroscopic characterization [52,53]. This constraint necessitates a comparative approach, where we draw analogies with the general vibrational behaviors of sulfur-based compounds, which are better studied in this spectral range. We postulate that 526 cm^−1^ and 536 cm^−1^ peaks correspond to the vibrational frequencies of Se-O and Te-O bonds, respectively. The larger atomic radius of Te compared to Se implies a stronger tendency for Te to engage in bonding interactions with surrounding atoms. This could be the reason for its heightened influence on structure and dynamics. The spectral region ranges from 638 cm^−1^ to 654 cm^−1^, and the data suggest a set of vibrational modes likely associated with [Bi_2_O_2_]^2+^ units within the lattice. Although these vibrational modes are intrinsically stable due to the structural rigidity, they appear to be perturbed to a degree by the presence of Se and Te, resulting in a discernible shift in their absorption frequencies.

The Raman and FTIR spectra offer comprehensive insights into the vibrational properties of Te-doped Bi_2_O_2_Se samples. The spectra reveal the coexistence of the Se and Te phases and highlight the subtle shifts in vibrational frequencies due to their mutual influence. These analyses suggest the significant impact of Te substitution on the vibrational dynamics of the compound, enhancing our understanding of its structural and dynamic intricacies.

Figure 4 and Appendix A present SEM images of Bi_2_O_2_Se_1–x_Te_x_ samples. The images reveal individual flake-like or platelet-shaped particles with irregular contours and varying degrees of aggregation (Figure 4a–c). These observations suggest heterogeneous crystalline growth. The morphology of Bi_2_O_2_Se_1–x_Te_x_ appears inherently layered, with discrete layers accumulating atop one another to form larger composite structures. An analysis across twenty randomly selected regions for each material type yielded an average particle size of 3.4 ± 0.7, 3.2 ± 0.9, 3.9 ± 1.0, 4.2 ± 0.8, and 2.6 ± 0.6 μm for various Te doping concentrations in Bi_2_O_2_Se_1–x_Te_x_ (x = 0, 0.2, 0.4, 0.7, 1). Accompanying each SEM image are schematic diagrams representing the crystal structures’ side-view, oriented to view from the [100] crystallographic plane. This perspective offers a clear view of the structure’s layer stacking and atom distribution. The transition from Bi_2_O_2_Se through Bi_2_O_2_Se_0.4_Te_0.6_ to Bi_2_O_2_Te is visualized, showing how Te substitution affects the microstructure. Under SEM, we cannot detect an obvious difference in structure or growth shape. As Te is substituted for Se, increases in layer spacing could be expected due to differences in atomic size and bonding preferences of Te and Se. XRD and TEM tests were performed on all the samples, and the results obtained from the crystal structure and lattice spacing confirmed our conjecture, which will be discussed in the next paragraph. Based on the observations, the schematic illustration in Figure 4d abstracts the generic layered architecture typical to the Bi_2_O_2_Se_1–x_Te_x_ system, emphasizing the repeating nature of the layers.

Based on Figure 4a–c and the existing Bi_2_O_2_Se structure diagram, we have carefully drawn a diagram illustrating the continuous incorporation of Te into Bi_2_O_2_Se from top to bottom. This diagram clearly shows that as the amount of Te incorporation increases, the layer spacing from the bottom to the top also expands. There is an obvious positive correlation between this trend and the amount of Te incorporation, which provides intuitive and powerful evidence for understanding the structural properties of Bi_2_O_2_Se. We can find that the expansion of layer spacing is not obvious in the initial stage of Te incorporation, which may be because Te atoms have not yet significantly impacted the Bi_2_O_2_Se structure. However, as Te incorporation gradually increases, the layer spacing expansion trend becomes more significant. This change not only affects the overall shape of the Bi_2_O_2_Se structure but may also profoundly impact its physical and chemical properties. Additionally, when Te doping is further increased, the change in layer spacing gradually slows down due to the proximity to the Bi_2_O_2_Te structure. This may be because the distribution of Te atoms in the Bi_2_O_2_Se structure is gradually becoming saturated. This phenomenon provides a new idea and research direction for further exploring the properties of the Bi_2_O_2_Se structure.

Figure 5 and Appendix A display the HRTEM and SAED analyses of Bi_2_O_2_Se_0.4_Te_0.6_, Bi_2_O_2_Se, and Bi_2_O_2_Te. We chose the lamellar structures of Bi_2_O_2_Se_0.4_Te_0.6_, Bi_2_O_2_Se, and Bi_2_O_2_Te to examine their atomic arrangements and crystalline shapes, as shown in Figure 5a,d,g. Generally, their appearances are consistent with those of layered materials, indicative of stacking or similar growth mechanisms. Figure 5b,e,h reveal that the SAED patterns exhibit sharp, well-defined diffraction spots, confirming the single-crystalline nature of the samples. The symmetrical arrangement of these spots implies a high degree of crystallinity and order within the structures, indicating the absence of significant grain boundaries or polycrystalline domains in the examined areas. By calculating and comparing the crystal face spacing, we determined that all samples exhibit the (110), (110), and (020) structures and share the same crystal type [54,55]. This finding is vital for understanding the substitutional effects of these compounds. Additionally, we observed that the crystal pattern of Bi_2_O_2_Se_0.4_Te_0.6_ aligns with existing crystal patterns. The lattice spacings of Bi_2_O_2_Se_0.4_Te_0.6_, Bi_2_O_2_Se, and Bi_2_O_2_Te were calculated to be 0.275 nm, 0.28 nm, and 0.27 nm, respectively, using Density Functional Theory (DFT) and Inverse Fast Fourier Transform (IFFT). They are consistent with our predictions. Bi_2_O_2_Se_0.4_Te_0.6_ exhibited a smaller atomic spacing than Bi_2_O_2_Te, likely due to the effects of doping. This smaller spacing was observed consistently across multiple measurements. Such variations in lattice spacing can significantly influence various material properties, including band structure, mechanical strength, and thermal conductivity. Finally, the SAED pattern of Bi_2_O_2_Se_0.4_Te_0.6_ shows a more irregular atomic distribution compared to the SAED patterns of Bi_2_O_2_Te and Bi_2_O_2_Se due to the creation of excessive polycrystalline layer buildup and the presence of a considerable degree of lattice distortion in Bi_2_O_2_Se_0.4_Te_0.6_.

### 2.3. Optimizing the Thermoelectric Performance of the Bi_2_O_2_Se_1–x_Te_x_

In an investigation of the thermoelectric properties of Bi_2_O_2_Se_1−x_Te_x_ variants, the thermal conductivity and thermal diffusion coefficient were graphically represented as a function of temperature (Figure 6a and Appendix A). Considering the potential application of waste heat recovery in daily life, especially the importance of improving energy efficiency and reducing energy waste, we focused our tests on the temperature range from above room temperature to below 100 °C. This temperature range (30–100 °C) covers many sources of waste heat in domestic and industrial scenarios, such as water heaters, air conditioning systems, industrial cooling processes, etc. By efficiently recycling this waste heat, we can significantly reduce our dependence on fossil fuels while lowering greenhouse gas emissions.

To ensure the outstanding reliability of the investigated thermoelectric material in practical applications, it was subjected to exhaustive TG tests. The results revealed that even at extreme temperatures up to 600 °C (Appendix A), the sample’s mass remained virtually unchanged, with no significant loss or fluctuation. This finding is strong evidence that the thermoelectric material has excellent thermal stability and is capable of operating at sustained high temperatures for extended periods. Simultaneously, XRD tests were performed on the heated samples to further verify their structural stability. The results showed that the high-temperature-treated samples had identical XRD patterns to those before heating, meaning the 600 °C temperature did not affect their crystalline shape. This important discovery has far-reaching practical significance for waste heat recovery applications. It fully proves that the thermoelectric material is capable of continuously and effectively converting waste heat into electricity in various complex and changing practical environments. This undoubtedly provides broader prospects and possibilities for future energy conversion and utilization.

The data clearly reveal a phenomenon: the material’s thermal conductivity remains almost constant at different temperatures. However, the lattice thermal conductivity of the material exhibits a unique trend of decreasing and then increasing with the gradual increase of Te concentration. This change in thermal conductivity mainly stems from the point defects and mass inhomogeneities generated during the Te substitution process. Specifically, Te^2−^ ions possess greater mass and volume than the relatively lighter and smaller Se^2−^ ions. When these Te^2−^ ions enter the material’s lattice, they inevitably induce significant distortions. These lattice distortions exacerbate the phenomenon of phonon scattering, a key factor in the thermal conductivity of materials. Enhanced phonon scattering leads to a reduction in overall thermal conductivity, which partly explains why the lattice thermal conductivity initially tends to decrease as the Te concentration increases. This downward trend reaches a turning point when the doping reaches half the amount. At this point, the structure of the material begins to evolve toward a more regular, ordered Bi_2_O_2_Te structure. This structural change reduces the number of defects and increases the regularity of the lattice, which in turn helps to improve thermal conductivity. As a result, we observe an increasing trend in the lattice thermal conductivity after more than half of the doping amount. These empirical observations lay a crucial groundwork for fine-tuning the thermoelectric attributes of Bi_2_O_2_Se via Te doping [41]. 

To comprehensively characterize the thermoelectric properties of Bi_2_O_2_Se_1–x_Te_x_ samples, electrical resistivity measurements were conducted on specimens with a diameter of 12.7 mm and a thickness of 10 mm. Figure 6b provides critical insights into the resistance variation of the Te-doped samples. Between 30–100 °C, the resistance is significantly influenced by the Te doping amount. Notably, conductivity exhibits optimal performance when the Te doping lies between 0.4 and 0.8. Within this temperature range, conductivity steadily increases with rising temperature, potentially related to the thermal excitation of carriers and increased mobility. However, an intriguing phenomenon appears at higher temperatures: the resistance gap between doped and undoped samples narrows. This suggests that at elevated temperatures, the material’s intrinsic conductive mechanism dominates, weakening the doping effect on resistance.

The Bi_2_O_2_Se_0.4_Te_0.6_ sample exhibited critical resistivity behavior. At room temperature, its conductivity was 1.86 S/cm, but at 100 °C, it dramatically increased to 17.36 S/cm. A similar significant change was observed in the Bi_2_O_2_Se_0.2_Te_0.8_ sample, leading to the hypothesis that a sample with higher conductivity may exist between the two. This remarkable resistivity change is typical of semiconductor materials, where increasing temperature alters carrier concentration, affecting electrical conductivity. The sharp resistivity increase from room temperature to 100 °C indicates a significant impact on the carrier transport mechanism within the material. We decided to explore this possibility in-depth in a follow-up study. This remarkable change in resistivity is typical of the behavior of semiconductor materials. The resistivity of semiconductor materials usually increases with increasing temperature, which is due to the fact that an increase in temperature leads to a change in the carrier concentration within the material, which affects its electrical conductivity. When the temperature is increased from room temperature to 100 °C, the sharp increase in resistivity indicates that the carrier transport mechanism inside the material is significantly affected.

The temperature-dependent conductivity data were analyzed using the small polaron classical hopping mechanism, which showed excellent agreement with an Arrhenius-type equation. This confirms that thermally-activated hopping processes predominantly govern carrier transport in Te-doped Bi_2_O_2_Se samples. Such a theoretical framework cohesively rationalizes the observed electrical behaviors in our study [56]:(1)$$\sigma =\mathrm{ne}\mu =\left(\frac{C}{T}\right)\exp\left(\frac{-E_{a}}{k_{B}T}\right)$$

In the framework of electrical conductivity, the Arrhenius-type equation governs the temperature-dependent resistivity behavior, where *C* is the pre-exponential factor, *k_B_* is the Boltzmann constant, *E_a_* is the activation energy, and *T* is the absolute temperature. A robust linear correlation between *ln(rT)* and 1000/*T* substantiates this conductivity mechanism. According to this model, the activation energy *E_a_* for electron conduction diminishes as Te content in the samples rises. Specifically, *E_a_* values for Bi_2_O_2_Se, Bi_2_O_2_Se_0.4_Te_0.6_, and Bi_2_O_2_Se_0.2_Te_0.8_ are 0.33, 0.19, and 0.20 eV, respectively. This trend signifies that Te substitution makes the intrinsic excitation of electrons more energetically favorable, thereby enhancing electrical conductivity to some extent.

With increasing Te doping concentration in the Bi_2_O_2_Se system, we observe a remarkable phenomenon: the conductivity exhibits an increasing then decreasing trend, becoming more pronounced with higher temperatures and widening gaps. This behavior can be attributed to two main influencing factors: the increase of carrier concentration and the decrease of carrier mobility. Firstly, the increased carrier concentration stems from the successful substitution of Te ions for ions at the Se lattice positions. This substitution narrows the material’s band gap, allowing electrons to more easily jump from the valence to the conduction band, resulting in a substantial carrier (electron and hole) concentration increase of many orders of magnitude, reaching around 10^18^ cm^−3^. This huge increase effectively transforms the original Bi_2_O_2_Se into a robust n-type semiconductor exhibiting excellent electrical conductivity. However, we also observed unfavorable effects as the carrier concentration surged. The substitution of Te ions introduces new point defects, creating a strong scattering effect in the lattice that severely obstructs carrier motion. These point defects and enhanced carrier scattering impair electron mobility to some extent, limiting the efficiency and speed of electron motion within the material.

Therefore, although the increase in carrier concentration initially enhances conductivity, the decreasing mobility starts to dominate with further doping, leading to a declining conductivity trend. Te doping in Bi_2_O_2_Se triggers a series of complex physical processes, ranging from conductivity enhancement to mobility decrease. The interplay of these factors determines the complex behavior of material conductivity as a function of doping concentration and temperature.

Figure 6c delineates the Seebeck coefficients (*S*) for all Bi_2_O_2_Se_1−x_Te_x_ samples, exhibiting behavior typical of n-type semiconductors at room temperature. 

As the Seebeck coefficient is negatively correlated with carrier concentration, the absolute value of the Seebeck coefficient S for the corresponding samples decreases with increasing carrier concentration. This trend is clearly visualized in Figure 6c, where we can observe the value of S decreasing with increasing Te doping. This variation rule provides a possible way to optimize the material’s Seebeck coefficient by modulating the carrier concentration. By precisely controlling the Te doping concentration, we can expect to realize precise regulation of the material’s Seebeck coefficient, further expanding its application potential in fields such as thermoelectric conversion. Interestingly, no significant variance in the Seebeck coefficients is observed among the doped samples at elevated temperatures. This behavior is effectively described by the semiclassical Mott–Jones formula [55], which encapsulates the relationship between *S* and temperature.
(2)ST=−π2kB2m*3π223h2e1n23

The Seebeck coefficient is theoretically influenced by several parameters, including the effective mass of carriers (*m*^∗^), the Planck constant (*h*), the elementary charge (*e*), and carrier concentration (*n*). Samples with elevated carrier concentrations are expected to exhibit a more gradual temperature-dependent change in the Seebeck coefficient. Additionally, a narrower band gap enables easier electron transitions, thereby attenuating the rate of increase in the absolute value of *S*. Consequently, Te-substituted Bi_2_O_2_Se manifests comparable *S* values at elevated temperatures, a phenomenon also observed in Te-substituted BiCuSeO.

Turning to electrical transport properties, the ZT for all Bi_2_O_2_Se_1−x_Te_x_ samples was calculated, revealing a consistent positive correlation with temperature. The formation of solid solutions between Bi_2_O_2_Se and Bi_2_O_2_Te has resulted in notable improvements in electrical transport performance, attributable to enhanced electrical conductivity and a moderate reduction in the Seebeck coefficient. The data presented in Figure 6d on the temperature-dependent ZT values reveals a striking enhancement in the thermoelectric performance of Te-substituted Bi_2_O_2_Se samples. The peak ZT value for Bi_2_O_2_Se_0.4_Te_0.6_ reaches 0.86 at 373 K, a six-fold increase compared to pure Bi_2_O_2_Se. This increase is not trivial; it places Te-doped Bi_2_O_2_Se into a competitive category of thermoelectric materials. It is beneficial to compare these results with existing high-performing thermoelectric materials. For instance, Cl-doped SnSe, often cited for its high ZT values, reaches a ZT of approximately 1.5 at similar temperatures. While SnSe’s ZT can reach up to 4 at ultra-high temperatures, its performance is much more temperature-sensitive than Te-doped Bi_2_O_2_Se. This suggests that the latter could offer a more stable and reliable material for thermoelectric applications within the low thermoelectric material range below 100 °C. 

## 3. Conclusions

This study furnishes robust experimental evidence elucidating the substantial enhancements in electrical and thermal attributes achieved through substituting Te for Se in Bi_2_O_2_Se, thereby outperforming Bi_2_O_2_Te. X-ray diffraction (XRD) analysis corroborates that Te substitution does not adversely affect the crystalline structure, ensuring structural integrity. Exceptional thermoelectric performance is realized in the Bi_2_O_2_Se_0.6_Te_0.4_ configuration, where Te replaces 40% of Se atoms. Introducing heavier Te atoms as point defects effectively curtails the phonon mean free path, engendering a conspicuous reduction in lattice thermal conductivity. Concurrently, forming a solid solution with structurally analogous Bi_2_O_2_Te precipitates a notable band gap narrowing, culminating in augmented carrier concentrations and a significant attenuation in the activation energy required for electrical conduction. Consequently, this leads to a marked elevation in electrical conductivity and the ZT. However, the situation changes when the Te content becomes excessively high. At this point, the overall structure tends to shift toward the Bi_2_O_2_Te mode. Although this transformation does not bring about a destructive change in the crystal structure, it adversely affects the thermoelectric properties. This is because excessive Te content leads to high carrier concentration, which increases the electron-to-electron scattering probability and decreases electrical conductivity. Simultaneously, excessively high Te content may also cause excessive lattice distortion, leading to excessive phonon scattering and further reducing thermal conductivity. However, in thermoelectric materials, an equilibrium between electrical and thermal conductivities must be achieved to realize optimal thermoelectric performance.

The figure of merit (ZT) for Bi_2_O_2_Se_0.6_Te_0.4_ peaks at an impressive 0.86 at 373 K, setting a new benchmark for the Bi_2_O_2_Se material series. This investigation provides a comprehensive theoretical and empirical framework explaining the fundamental mechanisms underlying the improved thermoelectric performance achieved through Te substitution. The findings offer compelling empirical substantiation for the engineering potential of Te-substituted Bi_2_O_2_Se in practical applications, heralding significant advancements in developing high-performance thermoelectric materials suited for green and sustainable energy solutions.

Experiments and Methods

Materials: Bi (99.99%) and Se powders (99.99%) were purchased from Aladdin. The powder (99.99%) and Bi_2_O_3_ powders (99.99%) were purchased from Macklin. All materials involved are stable to air and water.

Solid phase synthesis of Bi_2_O_2_Se_1–x_Te_x_: The synthesis of bulk Bi_2_O_2_Se_1–x_Te_x_ was conducted by solid phase synthesis (x = 0, 0.25, 0.5, 0.75, 1). Bi_2_O_3_, Bi, Se, and Te powders were grounded in stoichiometric ratios, mixed, sealed, and evacuated in a glass ampoule. The ampoule was then placed into a muffle furnace (KSL-1200X, HEFEI KEJING, Hefei, China) and was heated to 650 °C at 5 °C/min and held for 15 h to follow the synthesis route (3). Then, the furnace was cooled to room temperature naturally. The powder obtained was grounded and compressed into a pellet under the pressure of 30 MPa using an infrared tablet press (YLJ-40TA, HEFEI KEJING, Hefei, China) at 30 Mpa for further characterization.
(3)$$\frac{2}{3}{Bi}_{2}O_{3}+\frac{2}{3}Bi+(1-x)Se+xTe\to {Bi}_{2}O_{2}{Se}_{1-x}{Te}_{x}$$

Characterization of Bi_2_O_2_Se_1−x_Te_x_: The samples were characterized using various techniques to investigate their properties. Morphological analysis was performed using cold-field-emission scanning electron microscopy (SEM, SU8010, Hitachi, Chiyoda, Japan). The phase structure was determined by X-ray diffraction (D/MAX-2550V; Rigaku Cu Ka radiation, Tokyo, Japan). Diffuse reflectance spectra were obtained on powder samples for optical analysis using a conventional spectrophotometer (Lambda 950; Perkin Elmer, Buckinghamshire, UK). The thermal stability of the sample was tested by the TGA test (TG 209 F1 Libra, NETZSCH, Hanau, Germany). Electrical conductivity (r) and Seebeck (*S*) coefficient measurements were carried out on bar specimens using homemade equipment. The thermal diffusivity (D) of the bulk specimens (diameter: 12.7 mm) was determined using the laser flash method (LFA-467; Netzsch, Selb, Germany). The actual density (ρ) was measured with the proper density meter (AccuPyc II; Micromeritics, America, Norcross, GA, USA). Specific heat capacity (*Cp*) was determined through the sapphire method using the Differential Scanning Calorimeter (DSC250; TA, Newcastle, DE, USA). Subsequently, the thermal conductivity (*j*) was calculated using the formula *j* = *DCpq*. X-ray photoelectron spectroscopy (XPS) was performed to understand its elemental composition using the Thermo SCIENTIFIC Nexsa instrument (Thermo Scientific, Waltham, MA, USA). Additionally, the band gap of the material was estimated.

## Figures and Tables

**Figure 1 materials-17-01509-f001:**
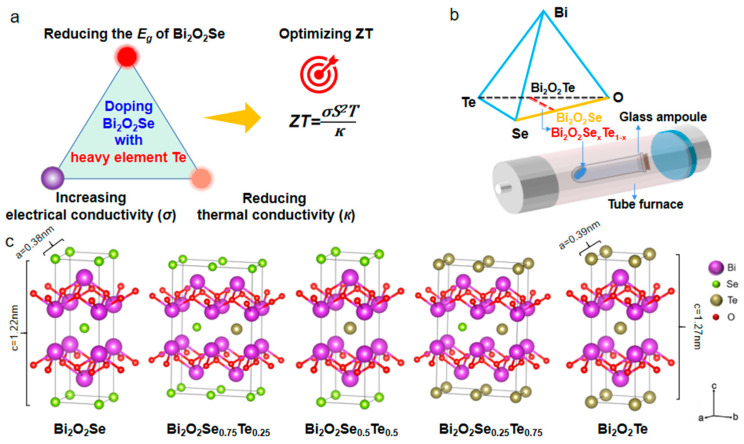
Optimization of thermoelectric properties through Te doping in Bi_2_O_2_Se crystals. (**a**) Schematic illustrating the strategic doping of Bi_2_O_2_Se with heavy Te atoms to enhance thermoelectric performance. (**b**) The quantitative phase diagram of the Bi_2_O_3_/Se/Te system and the solid-phase reaction methodology conducted in an evacuated glass ampoule demonstrate the compositional tunability. (**c**) Crystallographic representations of Bi_2_O_2_Se_1–x_Te_x_ compositions (where x = 0, 0.25, 0.5, 0.75, and 1), presenting the structural variations at distinct Te doping levels.

**Figure 2 materials-17-01509-f002:**
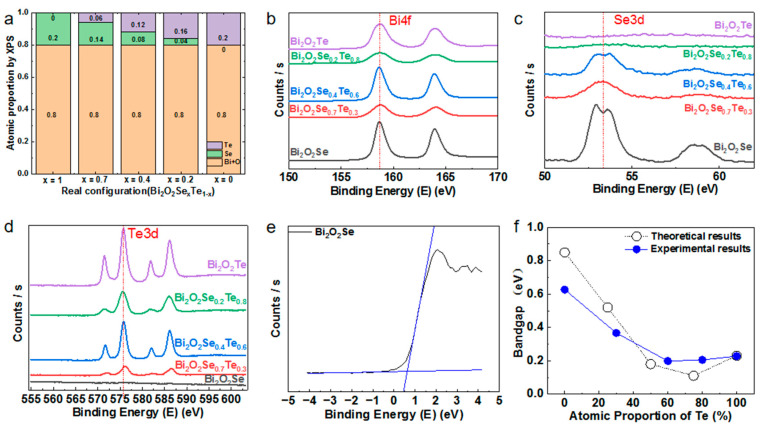
The XPS analysis of Bi_2_O_2_Se_1−x_Te_x_ at different Te doping levels. (**a**) The reaction result shows an actual configuration of Bi_2_O_2_Se_1−x_Te_x_ (x = 0, 0.2, 0.4, 0.7, 1). (**b**–**d**) XPS spectra for Se (**b**), Bi (**c**), and Te (**d**) of the as-grown Bi_2_O_2_Se_1−x_Te_x_ (x = 0, 0.2, 0.4, 0.7, 1) powders. (**e**) Method for estimating band gap width by valence band spectrum. (**f**) Band gap of Bi_2_O_2_Se_1–x_Te_x_ with the comparison of theoretical calculations with experimental results.

**Figure 3 materials-17-01509-f003:**
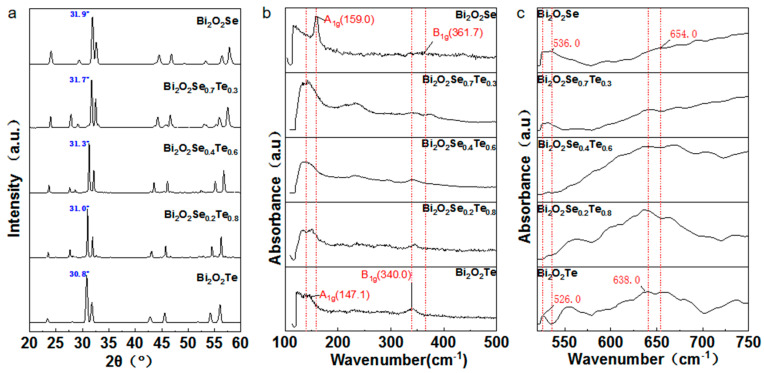
Comprehensive structural characterization of Te-doped Bi_2_O_2_Se. (**a**) X-ray diffraction patterns illustrate the systematic shift of peak angles towards lower values, indicating an increase in Te substitution concentration in Bi_2_O_2_Se_1−x_Te_x_. (**b**) Raman spectroscopic analysis shows the coexistence of multiple phases whose relative intensities are modulated by the Te-to-Se ratio in Bi_2_O_2_Se_1–x_Te_x_ samples. (**c**) The FTIR spectra acquired for bulk Bi_2_O_2_Se_1−x_Te_x_ samples contribute an additional layer of structural insight. The red lines are the types and locations of major peaks that we’ve marked.

**Figure 4 materials-17-01509-f004:**
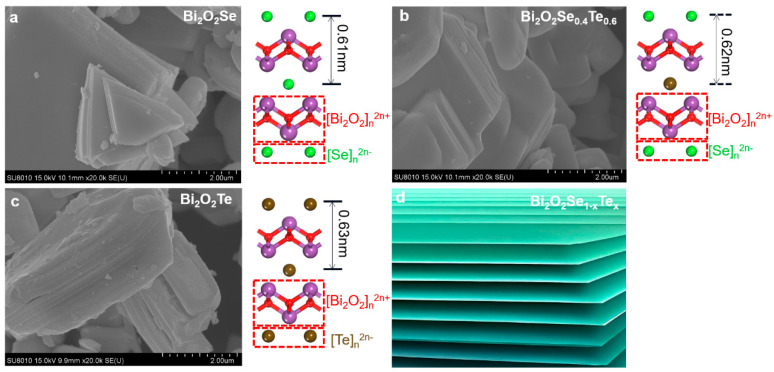
Microstructural analysis of representative Bi_2_O_2_Se_1–x_Te_x_ powders across different compositions. (**a**–**c**) The SEM images depict the layered structures of Bi_2_O_2_Se, Bi_2_O_2_Se_0.4_Te_0.6_, and Bi_2_O_2_Te. Accompanying schematic diagrams provide a side-view representation of the crystal structures for each composition, as viewed from the [100] crystallographic plane. (**d**) Schematic illustration highlighting the generic layered architecture of Bi_2_O_2_Se_1–x_Te_x_.

**Figure 5 materials-17-01509-f005:**
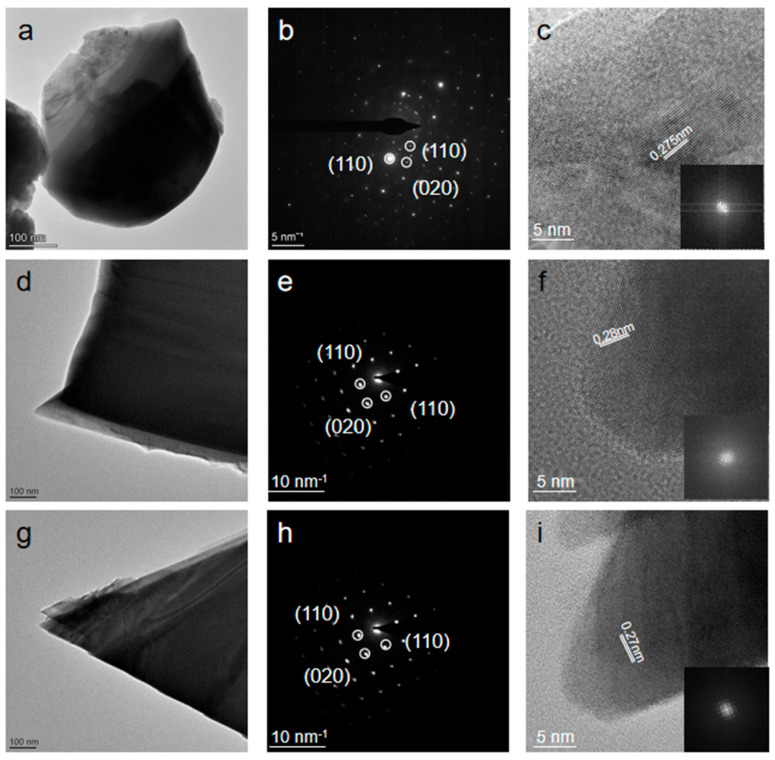
TEM characterization of Bi_2_O_2_Se_0.4_Te_0**.6**_ and Bi_2_O_2_Te and Bi_2_O_2_Se compounds. (**a**,**d**,**g**) Low-magnification TEM images presenting an overview of the microstructure for Bi_2_O_2_Se_0.4_Te_0.6_ (**a**), Bi_2_O_2_Te (**d**), and Bi_2_O_2_Se (**g**). (**b**,**e**,**h**) Selected area electron diffraction (SAED) patterns for Bi_2_O_2_Se_0.4_Te_0.6_ (**b**), Bi_2_O_2_Te (**e**), and Bi_2_O_2_Se (**h**) demonstrate their single-crystalline nature. (**c**,**f**,**i**) High-resolution TEM (HRTEM) images displaying the lattice spacings of 0.275 nm, 0.28 nm, and 0.27 nm for Bi_2_O_2_Se_0.4_Te_0.6_, Bi_2_O_2_Te, and Bi_2_O_2_Se.

**Figure 6 materials-17-01509-f006:**
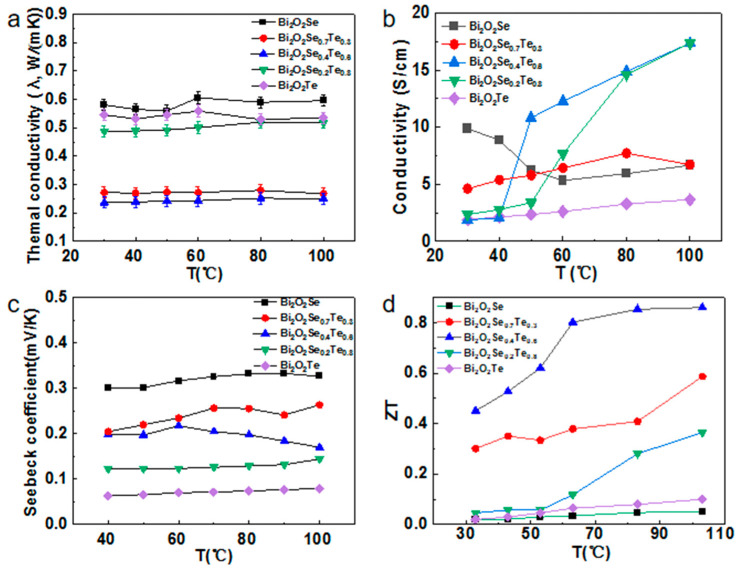
Optimization of thermoelectric properties in Bi_2_O_2_Se_1–x_Te_x_ compounds. (**a**) Graphical representation of the thermal conductivity across varying compositions of Bi_2_O_2_Se_1–x_Te_x_. (**b**) The inverse correlation between conductivity and temperature across different sample compositions. (**c**) Temperature-dependent Seebeck coefficient data for different compositions of Bi_2_O_2_Se_1–x_Te_x_. (**d**) The temperature-dependent figure of merit presents the optimized composition for thermoelectric performance as Bi_2_O_2_Se_0.4_Te_0.6_.

## Data Availability

Data are contained within the article and Appendix A.

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
