# Peer review of "Optimizing Thermoelectric Performance of Hybrid Crystals Bi2O2Se1−xTex in the Bi2O2X System"

_materials, 2024, doi:10.3390/ma17071509_

Round 1

Reviewer 1 Report

Comments and Suggestions for Authors

Big work done. Useful temperature range is chosen. Still the article should be polished to reach the publication stage. On the whole, deeper and thorough analysis of influence of X, from 0 to 1, is needed. Here are some notes to be corrected/answered:

1)       What is the real bandgap of Bi2O2Se: 0.63 eV (line 182) or 1.77 eV (line 187)?

2)       Big work, but what sense of giving size statistics (line 265) if there is no apparent tendency and conclusion?

3)       Fig. 4. What is the relation between a-c and d? May be SEM side-view of the layers with higher resolution could give more information about the repeating nature of the layers? Fig. d is not clear: does Te amount changes with vertical? Why the gaps between layers are smaller at the top?

4)       Fig. 5. SAEDs and HRTEMs of Bi2O2Se1-xTex with lower x values (x=0, 0.2,…) would give more evidence of Te impact on crystallinity and on atomic spacing.

5)       Line 327. Statement “the lattice thermal conductivity manifests a diminishing trend with an increasing concentration of Te in the material.” is not true. As Fig. 6a shows, 0.3 and 0.6 of Te really drops down the thermal conductivity, but 0.8 and 1.0 raises it back close to X=0. Analysis is needed.

6)       Fig. 6. Marking of specimen having the same amount of Te should be the same, in symbols and in color, in all a-b-c-d layers. Also, I strongly recommend to arrange the legends in ascending order of X, from top to bottom. Now it is very hard to watch the tendencies of X influence.

7)       Line 337. Statement “Figure 6b illustrates that Te-doped samples exhibit a marginally higher electrical resistance than undoped Bi2O2Se in the 30-100°C range” is not true. Deeper analysis of Fig.6b results needed.

8)       Line 340. The same statement “A key observation is the resistivity behavior of the Bi2O2Se0.4Te0.6sample, which shows a significant increase…” can be applied to X=0.8 sample.

9)       Line 342. “demonstrating a characteristic semiconductor behavior.” All samples, but X=0, demonstrate rise of conductivity with temperature, therefore sample X=0.6 is not a key exception.

10)   I recommend to show the Arrhenius plots of conductivity – it would evidence the declared activation energies (line 364).

11)   Line 367. No evidence of the statement “As Te concentration increases, an initial decline in electrical resistance is observed, stabilizing around room temperature.”

12)   Line 377. Fig. 6c does not agree with the statement “Given that the Seebeck coefficient is inversely related to carrier concentration, the augmented carrier concentration consequentially reduces the absolute values of S for all Te-substituted samples.” Not for all.

13)   Why thermoelectric properties drop down at high amount of Te?

Reviewer 2 Report

Comments and Suggestions for Authors

This manuscript reports on the optimization of the thermoelectric performance of a mixed crystals Bi2O2Se1–xTex in the Bi2O2Se/Bi2O2Te pseudobinary bulk system. The authors chose an interesting material system that attracts significant research interest in the context of 2D and pseudo-2D semiconductors. In addition, the employed characterization techniques (XRD, Raman, FTIR) are adequate for the purpose. The topic, as well as its physical and application-related implications, has a broad impact on different fields of research, particularly for novel thermoelectrical materials. The degree of novelty of the present study is high.

The methodology of HRTEM and SAED analyses also matches very well the envisaged studies and gives a solid background of the discussion. Thus, also the results obtained are not only credible but far reaching. In addition, results presentation is clear and easy to perceive helped by informative figures.

The present manuscript raises only minor technical concerns. These points amount to a minor revision before acceptance of this excellent manuscript for publication:

1: Title which is presently too long and may be perceived as unattractive to the readership, can be optimized and shortened in the interest of publicizing better the present work. It is incorrect in English to begin each word in a title with a capital letters.

2: The abstract should explicitly mention the characterization techniques employed.

3: The thermal stability the phases of interest should be discussed more explicitly, there are both experimental results and modeling pathways for testing energetics and for concluding on thermal stability of doped nanostructured semiconductor material systems.

4: In some paragraphs, refer to “temperature ranges” without explicitly stating the corresponding real or expected temperature ranges. Wherever possible, such expressions should be concreted, mentioning explicitly the temperatures.

5: The authors should also refer to the recent literature whereby modeling such as DFT and affordable ab initio molecular dynamics has been applied for looking at the thermal properties/stability and doping of 2D semiconductor materials, e.g., CrystEngComm 23 (2021) 6661-6667, and CrystEngComm 25 (2023) 5810-5817. Conceptually, this should be reflected in the introduction.

Comments on the Quality of English Language

The manuscript still needs a comprehensible grammatical revision.
